# The Progestin Medroxyprogesterone Acetate Affects HIV-1 Production in Human Lymphoid Tissue Explants in a Dose-Dependent and Glucocorticoid-like Fashion

**DOI:** 10.3390/v13112303

**Published:** 2021-11-18

**Authors:** Christophe Vanpouille, Gökçe Günaydın, Mattias Jangard, Mario Clerici, Leonid Margolis, Kristina Broliden, Andrea Introini

**Affiliations:** 1Section on Intercellular Interactions, Eunice Kennedy Shriver National Institute of Child Health and Human Development, National Institutes of Health, Bethesda, MD 20892, USA; vanpouic@mail.nih.gov (C.V.); margolil@mail.nih.gov (L.M.); 2Center for Molecular Medicine, Department of Medicine Solna, Division of Infectious Diseases, Karolinska University Hospital, Karolinska Institutet, 171 77 Solna, Sweden; gokcegnydnmbg@gmail.com (G.G.); kristina.broliden@ki.se (K.B.); 3Ear, Nose and Throat Unit, Research Laboratory, Sophiahemmet University, 114 86 Stockholm, Sweden; mattias.jangard@sophiahemmet.se; 4Department of Pathophysiology and Transplantation, University of Milan, 20122 Milan, Italy; mario.clerici@unimi.it; 5IRCCS Fondazione Don Carlo Gnocchi, 20148 Milan, Italy; 6Department of Biomedical and Clinical Sciences L. Sacco, University of Milan, 20157 Milan, Italy

**Keywords:** hormonal contraception, sex hormones, progesterone, DMPA, glucocorticoids, HIV-1, cytokines, lymphoid tissue, tonsils, tissue explants

## Abstract

The association between the use of the injectable contraceptive depot medroxyprogesterone acetate and HIV-1 susceptibility has been addressed mainly in respect to the changes occurring in the female genital mucosa and blood. However, one of the main sites of HIV-1 pathogenesis is lymphoid organs. To investigate the immunoregulatory effect of medroxyprogesterone acetate (MPA) at this site, human tonsillar tissue explants were infected ex vivo with either a CCR5 (BaL) or CXCR4 (LAI) HIV-1 variant and the release of p24_gag_ and cytokines was measured in culture supernatant. The response to MPA was compared with that elicited by treatment with progesterone (P4) and dexamethasone (DEX), which selectively binds the glucocorticoid receptor, in donor-matched explant cultures. MPA treatment reduced the replication of both tested HIV-1 strains as well as the production of the mediators of inflammation IL-1β, IL-17A and CCL5, but not CCL20, in a similar way to DEX, whereas P4 had no effect on HIV-1 replication. The magnitude of both MPA and DEX-mediated responses was proportional to the length of exposure and/or administered dose. Blockage of the progesterone and glucocorticoid receptors with mifepristone abolished all observed changes in HIV-1 and cytokine production, and was associated with increased IL-22 levels in HIV-infected explants. Our data indicate that elevated doses of MPA may affect the immune responses in lymphoid tissue in a glucocorticoid-like fashion with an immediate impact on local HIV-1 replication.

## 1. Introduction

Hormonal contraceptives (HC) account for more than 25% of all methods for family planning worldwide [1]. The number of injectable HC users increased more than four times in the past 25 years due to the advantages offered by inexpensive, short-acting contraceptives that are as effective as other methods with fewer administrations. For this reason, injectables represent the method of choice in resource-limited settings including Sub-Saharan Africa, where, with a prevalence of about 10%, injectables are the dominant contraceptive among women of reproductive age.

In spite of their higher efficacy compared to traditional barrier methods, HC do not protect against sexually transmitted infections (STIs). In particular, the administration of the progesterone (P4) synthetic analogue depot medroxyprogesterone acetate via intra-muscular injection (DMPA-IM) has been linked to increased risk of human immunodeficiency virus (HIV) 1 acquisition in a number of epidemiological, clinical and experimental studies over the years [2]. More recently, the ECHO study, a randomized trial conducted in Sub-Saharan Africa, did not find any difference in HIV-1 acquisition rate between users of DMPA-IM, a copper intrauterine device, and a progestin-based implant among women living in areas of high HIV incidence [3]. However, some limitations in study design, confounding behavioral factors and variability in experimental settings complicate the comparison and interpretation of all data produced to date, leaving unanswered important questions on the actual extent of the window of vulnerability and its modulation by natural and synthetic sex hormones [4].

Progestins such as medroxyprogesterone acetate (MPA) are designed to enhance drug bioavailability and bind the progesterone receptor (PR) with higher affinity than P4, although they are not selective for PR [5]. Of note, MPA binds to the glucocorticoid receptor (GR) with higher affinity than its natural ligand cortisol, triggering a dose-dependent anti-inflammatory response that has not been observed for P4 and other progestins [6,7]. Based on in vivo challenge experiments, it is well known that high doses of P4 or MPA affect both the physical and immune defenses of the female genital mucosa (FGM) promoting the establishment of a productive infection with STIs, including HIV, simian immunodeficiency virus, and herpes simplex viruses [8,9]. Likewise, functional experiments in humans evidenced an increased HIV-1 susceptibility and replication rate in FGM explants from women in the progesterone-dominant stage of the menstrual cycle [10] and upon MPA treatment ex vivo [11]. Nevertheless, the effect of systemic administration of MPA on organs other than the genital tract has been poorly investigated.

Lymphoid tissue is the main site of HIV-1 dissemination early after transmission and harbors the pool of latently infected cells, i.e., virus reservoir, which still represents the main obstacle to its eradication [12]. We investigated the effect of MPA on human tonsillar lymphoid tissue explants infected ex vivo with HIV-1, in comparison with the anti-inflammatory drug and selective GR agonist dexamethasone (DEX) as well as P4. Our results suggest that MPA, at doses that are compatible with peak concentrations observed in serum of DMPA users, exerts an immunoregulatory effect on lymphoid tissue comparable to that elicited by DEX, with implications that may extend to local immunity at mucosal sites of virus transmission.

## 2. Materials and Methods

### 2.1. Tonsillar Tissue Processing for Histoculture

Tonsils were immediately transferred upon surgery into jars containing phosphate buffer saline (PBS) for transportation. Specimens were kept at room temperature (RT) and processed within 4 h from surgery. Tonsils were dissected into tissue blocks, i.e., explants, according to a previously published procedure with minor adjustments [13]. Phenol red-free RPMI1640 medium (Thermo Fisher Scientific, Waltham MA, USA) and charcoal stripped fetal bovine serum (Thermo Fisher Scientific) at 15% final concentration (f.c.) were used to prepare culture medium. Tissue explants were cultured at the liquid-air interface on top of 1 × 1 cm pieces of gelatin sponge (Spongostan Standard 7 × 5 × 1 cm, Johnson & Johnson Medical N.V., Diegem, Belgium) in 4 mL of culture medium per well of 6-well culture plates. For each tissue donor, replicates were setup by allocating 9 explants to each sponge/well for a total of 18 explants per experimental condition. Explants were cultured for 12 days with a change of medium every 3 days. Cultures that showed clear sign of bacterial contamination, usually within Day 3 post-infection, were discarded.

### 2.2. Viruses

The following reagents were obtained through the NIH HIV Reagent Program, Division of AIDS, NIAID, NIH: human immunodeficiency virus-1 BaL, contributed by Dr. Suzanne Gartner, Dr. Mikulas Popovic and Dr. Robert Gallo [14]; human immunodeficiency virus-1 LAI, contributed by Dr. Jean-Marie Bechet and Dr. Luc Montagnier [15]. A cell-free virus stock was produced in human peripheral blood mononuclear cells. Culture supernatant was harvested at Day 10 post-infection, passed through a 0.45 μm filter, aliquoted and stored at −80 °C.

### 2.3. Compound Reconstitution

Medroxyprogesterone 17-acetate (MPA), mifepristone (RU-486), dexamethasone (DEX) and progesterone (P4) were purchased from Sigma-Aldrich (St. Louis MO, USA). All compounds were reconstituted in absolute ethanol 200 proof (Thermo Fisher Scientific) to make 1 mM solutions. Compound solutions were stored in sterile glass containers at −20 °C in the dark for up to 3 months. Compound solutions at 1000× the indicated assay concentrations for explant treatment were prepared in absolute ethanol and stored at −20 °C in the dark for up to 2 weeks, along with an aliquot of absolute ethanol to be used as vehicle control (0.1% f.c.).

### 2.4. Compound Treatment and Infection of Tissue Explants

Tissue explants were treated by adding 1000× compound solutions into culture medium and cultured overnight. Culture medium was replaced with fresh medium containing MPA or DEX at 1 nM f.c. or ethanol 0.1% f.c. (vehicle, VEH) before infection with HIV-1 and every 3 days up to 12 days (pre-infection setup). RU-486 was used at 1 µM f.c. before infection and throughout culture. In a separate set of experiments, MPA and DEX were used at the same concentrations from explant pre-treatment before infection throughout culture (before and after setup).

Explants were infected by delivering a volume of virus preparation of 5 to 7 µL containing the selected virus inoculum on top of each explant according to a previously published procedure [16]. The virus inoculum was empirically determined to achieve comparable virus production levels in tonsillar tissue explants and the volume of virus stock adjusted to deliver approximately 2000 and 500 pg of HIV-1 p24_gag_ in the selected volume of 5–7 µL for BaL and LAI respectively.

### 2.5. HIV-1 Quantification and Multiplex Cytokine Immunoassay

An aliquot of culture medium was harvested every 3 days before changing medium. Medium from replicate wells was pooled at collection for each time point, i.e., Day 3, 6, 9 and 12 post-infection. A bead-array immunoassay was used to quantify the amount of HIV-1 p24_gag_ and cytokines in culture supernatant according to a previously published procedure [17]. Values of p24_gag_ concentration in supernatant collected at the first change of medium, i.e., Day 3 post-infection, were excluded from the analysis because they are representative of the virus inoculum rather than new virus production.

For cytokine measurements, all monoclonal capture antibodies, biotinylated polyclonal detection antibodies, and human recombinant cytokines were purchased from R&D Systems (Minneapolis, MN, USA). Individual magnetic carboxylated bead sets (Luminex, Austin, TX, USA) were coupled to the capture antibodies according to the manufacturer’s recommendations. Briefly, human recombinant cytokines were resuspended at concentrations ranging from 6 to 100 ng/mL, and 1:3 serially diluted to generate a 10-point standard curve. Assays were run using 2000 beads per bead set in a total volume of 50 μL per well. Samples were run in duplicates at 2 dilutions 1:1 and 1:9. An aliquot of 50 μL of diluted sample was added to the bead mixture and incubated overnight at 4 °C in a Bio-Plex Pro flat bottom 96-well plate (Bio-Rad, Hercules, CA, USA). Standards and samples were diluted in assay buffer (20 mM Tris-HCl, 1% each normal mouse and goat serum (Gemini Bioproducts, West Sacramento, CA, USA) and 0.05% Tween-20 in PBS). Plates were washed twice with a solution of 0.05% Tween-20 in PBS. The beads were incubated with the detection antibody mixture for 60 min at RT. Biotinylated detection antibodies were used at twice the concentrations for a classic enzyme-linked immunosorbent assay recommended by the manufacturer. Plates were washed twice, and the beads incubated with a solution containing phycoerythrin-conjugated streptavidin 12 μg/mL (Thermo Fisher Scientific) for 30 min at RT. Beads were acquired with a Bio-Plex200 system (Bio-Rad). The median fluorescence intensity of a minimum of 100 beads per each bead set was recorded for each sample, and analyzed with the Bio-Plex Manager software (Bio-Rad) using a 5P regression algorithm. Concentration values that were below the lower limit of quantification (LLOQ) were reported as the midpoint value between zero and the LLOQ for statistical analysis. The LLOQ (pg/mL) was 8.2 for interleukin (IL) 1β, 27.4 for IL-17A, 45.7 for IL-22, 8.2 for C-C Motif Chemokine Ligand (CCL) 3, 8.2 for CCL4, 3.0 for CCL5, 22.9 for CCL20.

### 2.6. Statistical Analysis

All experiments were conducted independently using donor-matched tissue explants. To minimize the effect of biological variability between tissue donors, the results of cumulative HIV-1 p24_gag_ and cytokine production by treated explants (MPA, DEX, P4, and RU-486) were normalized to those of donor-matched untreated control explants (VEH), and the ratio of increase or decrease referred to as n-fold. N-fold values were log10 transformed and tested against 0 using the one sample *t* test. Differences in log-transformed n-fold values between two treatment groups were evaluated using a paired or unpaired *t* test (two-tailed) to compare results generated using explants from the same or an independent group of donors respectively (pre-infection vs before and after). Differences in n-fold change of cumulative HIV-1 p24_gag_ and cytokine production between multiple treatment groups were evaluated using the two-way ANOVA test with Sidak’s multiple comparisons test. *p* < 0.05 indicated statistical significance. All analyses were preformed using GraphPad Prism v8.4.3 (GraphPad Software, San Diego, CA, USA).

## 3. Results

### 3.1. Tissue Donors

Specimens of palatine tonsils were collected at the Sophiahemmet hospital from 14 children (8 females; 5 males; 1 information not available; age ≤ 10 years). All specimens were treated according to the experimental setup referred to as pre-infection.

Specimens of palatine tonsils were collected at the Children’s National Medical Center from 6 donors of unknown sex and age. All specimens were treated according to the experimental setup referred to as before and after (infection).

The reason for surgery was not available for any tissue donor.

### 3.2. MPA Treatment Reduces HIV-1 Production in a Dose-Dependent Fashion

To model the pharmacokinetic observed in plasma of DMPA users, characterized by declining levels of MPA after reaching peak concentrations, we incubated donor-matched tonsillar tissue explants with MPA at physiologically relevant concentrations, namely 10 and 100 nM, prior to HIV-1 infection and with MPA at 1 nM thereafter until the end of culture (Figure 1). Explant infection was carried out using a CCR5 and CXCR4 HIV-1 variant, namely BaL and LAI, to account for potential differences in compound effect on virus as well as cytokine production associated with target cell tropism [18].

Treatment of HIV-infected tissues with MPA 100 nM resulted in a reduction of cumulative virus production over culture time of an average 0.71- and 0.70-fold compared to untreated explants (VEH) for BaL (n = 6, *p* = 0.0033) and LAI (n = 8, *p* = 0.0108) respectively, whereas no statistically significant change was observed for MPA 10 nM (Figure 1B). Donor-matched tissue explants treated with the GR selective agonist dexamethasone (DEX 10 nM) yielded a decrease in virus production of an average 0.38- and 0.55-fold compared to VEH for BaL (n = 6, *p* = 0.0018) and LAI (n = 8, *p* = 0.0003) respectively. The extent of the changes associated with MPA and DEX treatment was not significantly different between BaL- and LAI-infected explants (Appendix A). Likewise, there was no difference in the extent of HIV-1 modulation by DEX and MPA between tissue explants from male and female donors (Appendix A).

The inhibitory effect on HIV-1 production was not observed neither in MPA- nor in DEX-treated explants when the GR and PR were blocked using mifepristone (RU-486 1 μM) in both BaL- and LAI-infected explants. In addition, explant treatment with either RU-486 alone or P4 100 nM did not result in any change in virus production. A statistically significant difference in HIV-1 production between MPA 100 nM, DEX 10 nM and the respective RU-486-treated groups was confirmed upon adjustment for multiple comparison in a pooled analysis of the results of LAI and BaL infection experiments (Appendix A).

In a second set of experiments, MPA and DEX were added to explant culture supernatant throughout culture time at the same concentration used for pre-treatment before infection, namely 10 and 100 nM (Figure 2A). The results of infection experiments with BaL (n = 2) and LAI (n = 4) were pooled according to the time of sample collection for statistical analysis due to the small sample size. As opposed to a shorter treatment, incubation with MPA 10 nM was associated with a statistically significant reduction of an average 0.76-fold in virus production compared to VEH (*p* = 0.0151, n = 6). This difference was more pronounced for treatment with MPA 100 nM and DEX 10 nM that resulted in an average 0.51- and 0.23-fold reduction in HIV-1 production respectively compared to VEH (*p* = 0.0007, n = 6). As observed in our first set of experiments, a longer incubation with P4 at 100 nM did not affect virus production. A statistically significant difference in HIV-1 production between MPA 10, 100 nM, DEX 10 nM and the respective RU-486-treated groups was confirmed upon adjustment for multiple comparison (Appendix A).

A pairwise comparison of MPA-treated groups between the two compound treatment setups (pre-infection vs before and after) demonstrated a statistically significant difference in the extent of HIV-1 reduction for both MPA 10 and 100 nM (*p* = 0.0071 and *p* = 0.0260 respectively, n = 6–14) (Figure 2B). In summary, the magnitude of the effect of MPA treatment on HIV-1 production in lymphoid tissue depended not only on the administered dose prior to infection, but also on the length of treatment thereafter during culture.

### 3.3. MPA Treatment Affects Cytokine Production in a Glucocorticoid-Like Fashion

To address the relationship between the observed effect on HIV-1 infection and tissue inflammation/cell activation status in tonsillar tissue explants, we measured the levels of selected pro-inflammatory cytokines and chemokines that have been previously shown to be modulated in blood and cervico-vaginal specimens of DMPA users or upon in vitro MPA treatment, namely IL-1β, IL-17A, IL-22, CCL3, CCL4, CCL5, and CCL20 [19]. We carried out this analysis on samples of culture supernatant from the same tissue explants that were treated with compounds and infected with HIV-1 as shown in Figure 1 (pre-infection, n = 5 BaL and n = 7 LAI) and Figure 2 (before and after, n = 1–2 BaL and n = 4 LAI). 

In tissues treated prior to infection, baseline production of IL-1β, IL-22, CCL3, CCL4 and CCL5 varied between BaL- and LAI-infected explants, possibly reflecting virus difference in cell tropism and subsequent CD4 T cell depletion (Appendix A). However, compound treatment resulted in comparable changes in cytokine production between the two HIV-1 strains (Figure 3A and Figure 4A). This effect was particularly evident for the potent anti-inflammatory drug DEX and characterized by a reduction of the pro-inflammatory cytokines IL-1β and IL-17A, in addition to IL-22 and the CCR5 ligands CCL4 and CCL5. Although IL-17A and CCL5 levels were the only cytokines that significantly differed from untreated control in both BaL- and LAI-infected explants, a significant difference between DEX and its RU-486 control was observed also for IL-1β and IL-22 in a pooled analysis of the results of LAI and BaL infection experiments upon adjustment for multiple comparison (pre-infection, Figure 3C and Figure 4C). Of note, DEX treatment was associated with higher production of CCL20 (Figure 3A), although this change was statistically significant for BaL but not LAI and did not differ from the RU-486 control (Figure 3C). As expected, the magnitude of the change in cytokine levels was enhanced by a longer treatment with DEX 10 nM (before and after). As for the analysis of p24_gag_ production, cytokine results from LAI and BaL infection experiments were pooled for statistical analysis due to the small sample size. DEX treatment throughout culture time resulted in a statistically significant reduction in all measured cytokines, including CCL3 and CCL4, and a significant upregulation of CCL20 (Figure 3B and Figure 4B). These changes were confirmed by the comparison between DEX and its RU-486 control that resulted in a statistically significant difference for all measured cytokines (before and after, Figure 3C and Figure 4C).

MPA elicited a similar response to DEX, although the extent of cytokine changes compared to untreated control was small in tissue explants treated prior to infection and the statistical significance of these changes inconsistent between BaL- and LAI-infected explants, with the exception of CCL5 that was significantly reduced by both MPA 100 and 10 nM for both HIV-1 strains (Figure 3A and Figure 4A). However, a longer incubation with MPA 100 nM resulted in a statistically significant reduction of IL-1β, IL-17A, IL-22, CCL3, CCL4 and CCL5 (Figure 3B and Figure 4B), which was confirmed by the comparison with its RU-486 control (before and after, Figure 3C and Figure 4C). Although there was only a trend towards higher CCL20 levels compared to untreated control (Figure 3B), CCL20 levels were significantly different between both MPA 100 and 10 nM and the respective RU-486 control (before and after, Figure 3C).

Of note, blockage of GR and PR by RU-486 was associated with higher levels of IL-22 than in untreated control explants irrespectively of the MPA dose administered prior to infection (Figure 3A), resulting in a statistically significant difference between both MPA 100 and 10 nM and the related RU-486 control (pre-infection, Figure 3C). While MPA 10 nM did not significantly affect baseline cytokine production regardless of the length of incubation, treatment with MPA 10 nM throughout culture time abolished the positive effect of RU-486 on IL-22 production (Figure 3B). Therefore, no difference in IL-22 levels was observed between MPA 10 nM and its RU-486 control (before and after, Figure 3C). In line with this observation, a longer treatment with MPA 100 nM and DEX 10 nM in combination with RU-486 also normalized IL-22 levels to baseline (Figure 3B), while the statistically significant difference observed between these two groups and the related RU-486 controls (before and after, Figure 3C) was likely due to a stronger GR engagement by treatment with a higher dose of MPA and a more potent GR agonist respectively, which resulted in IL-22 downregulation.

In summary, in agreement with the results on HIV-1 replication, the modulation of cytokine production in HIV-infected explants by MPA treatment was proportional to the administered dose as well as the length of treatment. Our data suggest that physiological doses of MPA may affect the immune responses in lymphoid tissue in a glucocorticoid-like fashion with an immediate impact on local HIV-1 pathogenesis.

## 4. Discussion

Mixed results have been produced to substantiate the observation that the use of DMPA increases women’s susceptibility to HIV [19]. In addition to blood and its derivatives, most clinical and experimental studies conducted to date have focused on the female genital mucosa (FGM), which is considered to be the main site of HIV acquisition in DMPA users. Although mucosal tissues are the port of entry of HIV and other STIs, viruses can rapidly spread to multiple organs to establish a productive infection that may ultimately result in a chronic and possibly life-long disease. Lymphoid tissue is not only the major site of HIV-1 replication early on during infection, it also harbors the majority of latently infected cells that support new virus production in the absence of effective immunological and/or pharmacological suppression [20].

The present study aims to address the potential effect of systemic administration of the progestin MPA on HIV-1 infection in lymphoid tissue. To this end, we employed human tonsillar tissue explants as an experimental platform that was previously demonstrated to recapitulate some key aspect of early HIV-1 pathogenesis in humans [21]. In our initial experiments we exclusively used specimens from children below 10 years old to eliminate any potential interference from different types and levels of endogenous sex hormones [22]. For the same reason, we used phenol red-free medium and charcoal stripped serum to minimize the effect of exogenous factors that could bind nuclear receptors thus confounding our analysis. Our experiments were designed to study the local effect of peak concentrations (Cmax) of MPA recorded in plasma upon administration of 150 mg DMPA-IM every 3 months. This value varies considerably between reports and falls within a range of 3 to 100 nM (1.15–38.5 ng/mL) [8]. Peak concentration is achieved within 3 weeks from administration and MPA plasma levels slowly decline reaching a plateau around 2.6 nM (1 ng/mL). Therefore, we selected concentrations of MPA representative of the top and average Cmax, namely 100 and 10 nM, as well as 1 nM as plateau concentration. Due to the impossibility of quantifying HIV exposure rate and pin point the time of transmission in vivo, we hypothesized that, if any, the extent of MPA effect on HIV susceptibility would be maximum around the peak concentration stage, and thus incubated explants with MPA prior to infection. Following this short pre-incubation, we modelled two different scenarios where, at first, we reduced the amount of compound to recapitulate declining levels of MPA observed in plasma of DMPA users upon reaching peak concentration (pre-infection). In a second set of experiments performed with tissues from different donors, we extended explant treatment with the same amount of compound used for treatment prior to infection throughout culture time to model a more sustained exposure over time (before and after).

As expected, the extent of the effect on HIV-1 infection was proportional to the administered dose of MPA and length of treatment of tissue explants. The inhibitory nature of such effect was compatible with the anti-inflammatory response elicited by treatment with DEX, a selective and potent GR agonist, and it was effectively reverted by blockage of both GR and PR with RU-486. In support of this observation, treatment with an elevated dose of the natural PR ligand P4 (100 nM) throughout culture did not affect HIV-1 replication as opposed to a 10-fold lower dose of MPA. However, we did not carry out cytokine analysis for P4-treated explants. In addition, the unavailability of selective PR and GR antagonists and the absence of an additional treatment group using a progestin with low affinity for GR, e.g., norethisterone, prevents us from excluding a synergistic effect via engagement of both receptors by MPA. The discrepancy between our results and those from the few studies of MPA effect on HIV-1 infection in human primary cells may depend on the different nature of tissues as well as experimental settings. An increase in HIV-1 replication associated with MPA in vitro-treatment was reported in peripheral blood mononuclear cells [23] and cervico-vaginal tissue explants [11] infected ex vivo with HIV-1. Although we did not exogenously stimulate explants to promote productive infection, in our system HIV-1 replication may be supported by endogenous production of mediators of inflammation induced upon mechanical damage during tissue dissection and/or hypoxia in culture. However, we consider this aspect a common limitation to all explant culture systems, and a direct impact of MPA on HIV-1 replication in a semi-polarized system of human ectocervical tissue explants was deemed minimal by an independent study [24]. Of note, a recent work on HIV-1 infection of humanized mice showed lower viral titers in vaginal secretions, but not in plasma, early after DMPA administration in spite of mice’s enhanced susceptibility to infection compared to the untreated control group [25]. The authors speculate that the observed transient reduction in HIV-1 local release may depend on the anti-inflammatory nature of MPA treatment in doses compatible with the Cmax in plasma of DMPA users, owing to its high affinity for GR.

While generally regarded as immunosuppressive, the net effect of MPA within the same individual is likely to vary with its concentration as highlighted by contrasting reports on MPA ability to affect cytokine levels, among other factors, in vaginal secretions and blood of DMPA users [19]. As opposed to GR that is ubiquitously expressed among human cells, contrasting reports exist on nuclear PR expression in immune cells [26], and to the best of our knowledge there is no evidence of its presence in tonsils. However, it cannot be excluded that PR and GR function is sexually dimorphic and varies with endogenous and environmental factors. Moreover, expression of non-classical PR on the cell surface was reported in immune cells, although their functional role in the immune-regulatory response elicited by P4 treatment has yet to be defined [27]. In our study, we did not observe any difference in the changes of HIV-1 production upon explant treatment with MPA, DEX and P4 between female and male tissue donors among children below 10 years old. While highly reproducible due to reduced interference from potential confounders, our results require further validation by using specimens from donors more representative of DMPA users. However, a faithful recapitulation of key aspects in the response to pharmacological treatment, e.g., genetics, co-infections and microbiota composition, could only be achieved by accessing samples in relevant geographic areas [28].

Even with less interference from inter-individual variability affecting population and clinical studies, we were able to measure a consistent downregulation of pro-inflammatory cytokines only in explants treated with the highest MPA dose (100 nM) and for the entire time of culture. In comparison with MPA, DEX was equally effective in reducing the same cytokines at a lower concentration (10 nM) and for a shorter incubation time. Of note, the chemokine CCL20 was the only factor among those measured that was increased by DEX as well as MPA. To our knowledge, only one study addressed the role of CCL20 in the context of MPA use, reporting increased gene expression levels upon in vitro treatment of primary human endometrial epithelial cells [29]. The modulation of antimicrobial factors with chemotactic function, such as CCL20, may account for the complexity of MPA action, resulting at the same time in a direct antiviral effect [30] and the recruitment of potential HIV target cells, such as Langerhans cells and Th17 cells, which express the CCL20 cognate receptor CCR6 [31,32]. However, some of the observed changes in cytokine levels might be partially dependent on HIV-1 infection and would need to be validated in uninfected tissue. In particular, the changes in IL-22 levels upon blockage of GR and PR by RU-486, and related modulation by MPA treatment, deserve further investigation due to the important role of this cytokine in the maintenance of epithelial barrier at mucosal sites and local antimicrobial responses [33,34]. In fact, one of the most consistently reported effects of DMPA use in humans as well as animal models is a thinning of the epithelial layer of the lower female genital tract, accompanied by a loss of tight junction and increased permeability of the mucosa, eventually resulting in higher susceptibility to HIV-1, among other STIs [35,36,37]. The molecular mechanisms underlying these changes have yet to be identified and, to the best of our knowledge no study has addressed the effect of MPA on IL-22 expression. In vitro treatment with GR agonists reportedly suppress cytokine expression in two of the main IL-22 producing cell subsets, namely T helper (Th) 17 and group 3 innate lymphoid cells (ILCs), via downregulation of key stimulators of its production, such as IL-1β and IL-23, as well as by promoting cell death [38,39,40]. In line with these data, MPA was reported to exacerbate HIV-1-associated CD4+ T cell depletion by inducing their apoptosis in a glucocorticoid-like fashion [41]. Of note, IL-22 and IL-17 producing CD4 T cells are among the main targets of the virus at mucosal sites of transmission and preferentially depleted early during HIV-1 pathogenesis [32,42,43]. We found that IL-1β and IL-22 levels were lower in HIV-1_BaL_- than HIV-1_LAI_-infected explants, possibly mirroring the selective cell depletion that occurs in vivo being transmitted-founder viruses exclusively CCR5-tropic [44]. However, increased levels of IL-22 in RU-486-treated explants was not accompanied by an increase in HIV-1 replication compared to untreated control possibly reflecting bystander T cell rescue from death and/or blockage of endogenous GR agonists such as cortisol. 

In conclusion, our data support the notion that MPA has an immunosuppressive effect on lymphoid tissue that is likely mediated by GR engagement. Although the dose-dependency of this effect is clear, we cannot exclude that a transient reduction of the local immune defenses upon MPA administration in vivo, possibly involving cell apoptosis, is followed by inflammatory processes such as cell recruitment and activation, including CCR6- and CCR5-expressing HIV-1 target cells [45]. While attempting to mirror declining plasma levels of MPA upon injection, our model cannot recapitulate any potential long-term changes nor the impact of a prolonged use of MPA following multiple injections [46], owing to intrinsic limitations in tissue explant integrity/viability over time [21]. Multiple cell subsets and soluble factors likely contribute to the observed structural changes in the FGM of DMPA users, including stromal and epithelial cells and related growth factors [25,47]. Future experiments should address the immunoregulatory effect of MPA, also in comparison with other progestins with lower affinity for GR, in lymphoid and mucosal human tissues, as well as tissue-derived primary cultures of different cell types, in the absence of HIV-1. Such investigation may lead to a better understanding of the mechanisms behind the local changes occurring at the main sites of transmission and pathogenesis of HIV-1 and other STIs, thus improving the implementation of safe hormonal contraception.

## Figures and Tables

**Figure 1 viruses-13-02303-f001:**
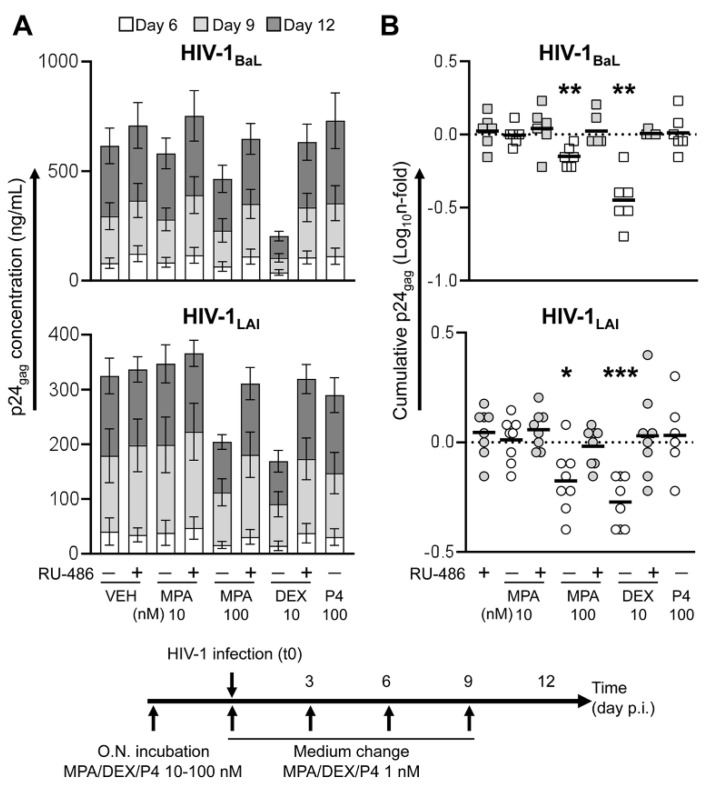
Virus production in tonsillar tissue explants treated with steroid hormones prior to HIV-1 infection (pre-infection). Tissue explants were incubated with medroxyprogesterone acetate (MPA), dexamethasone (DEX) and progesterone (P4) at the indicated concentrations, in the presence or the absence of mifepristone (RU-486) at 1 μM, overnight upon dissection. Culture supernatant was replaced with fresh medium containing MPA, DEX or P4 at 1 nM, in the presence or the absence of RU-486 at 1 μM, immediately before infection with HIV-1_BaL_ (top, n = 6) and HIV-1_LAI_ (bottom, n = 6–8). Explants were cultured using the same medium composition for 12 days with a change of medium every 3 days. (**A**) Kinetics of HIV-1 infection measured as the amount of p24_gag_ released in culture supernatant over time. Bars indicate mean values with s.e.m. (**B**) N-fold change in cumulative virus production (Day 6 to Day 12 post-infection) in supernatant of treated explants compared to donor-matched untreated explants (VEH). Lines indicate mean values. Asterisks denote a statistically significant difference with VEH (one sample *t* test,* *p* < 0.05, ** *p* < 0.01, *** *p* < 0.001).

**Figure 2 viruses-13-02303-f002:**
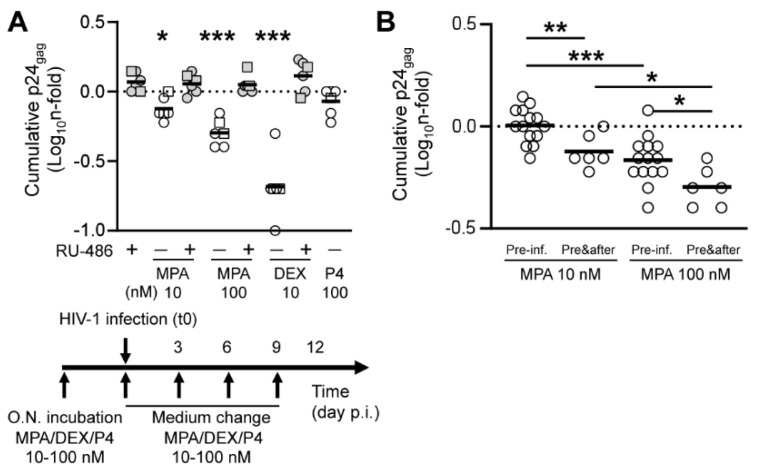
Virus production in tonsillar tissue explants treated with steroid hormones. (**A**) Virus production in tonsillar tissue explants treated with the same concentration of steroid hormones before and after HIV-1 infection (before and after). Tissue explants were incubated with medroxyprogesterone acetate (MPA), dexamethasone (DEX) and progesterone (P4) at the indicated concentrations, in the presence or the absence of mifepristone (RU-486) at 1 μM, overnight upon dissection. Culture supernatant was replaced with fresh medium containing the same amount of compound used for explant pre-treatment, in the presence or the absence of RU-486 at 1 μM, immediately before infection with HIV-1_BaL_ (squares, n = 2) and HIV-1_LAI_ (dots, n = 4). Explants were cultured using the same medium composition for 12 days with a change of medium every 3 days. The chart represents n-fold change in cumulative virus production (Day 6 to Day 12 post-infection) in supernatant of treated explants compared to donor-matched untreated explants (VEH). Lines indicate mean values. Asterisks denote a statistically significant difference with VEH (one sample *t* test,* *p* < 0.05, ** *p* < 0.01, *** *p* < 0.001). (**B**) N-fold change in cumulative virus production (Day 6 to Day 12 post-infection) in supernatant of explants treated with the indicated MPA concentration prior to infection (pre-inf., n = 14) or before and after infection (pre&after, n = 6) compared to donor-matched untreated explants (VEH). Lines indicate mean values. Asterisks denote a statistically significant difference between two groups (*t* test,* *p* < 0.05, ** *p* < 0.01, *** *p* < 0.001).

**Figure 3 viruses-13-02303-f003:**
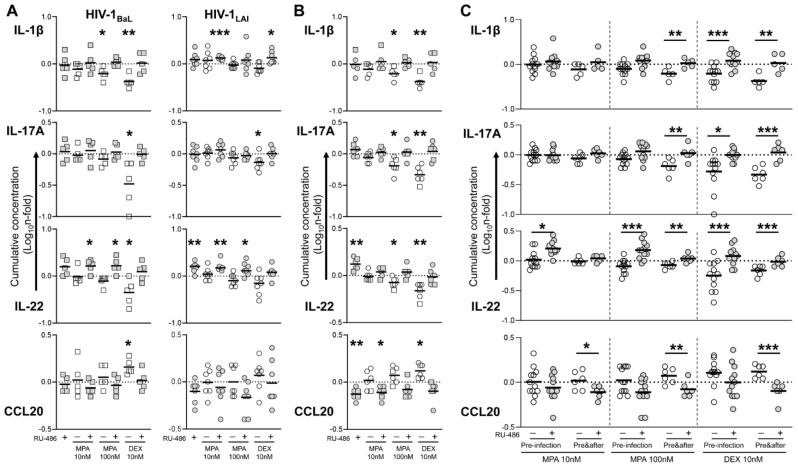
Cytokine production in HIV-infected tonsillar tissue explants treated with steroid hormones. Cytokine levels were measured in culture supernatant of tonsillar tissue explants treated with medroxyprogesterone acetate (MPA) and dexamethasone (DEX) at the indicated concentrations, in the presence or the absence of mifepristone (RU-486) at 1 μM, prior to infection (pre-infection, (**A**)), or before and after (pre&after, (**B**)) infection with HIV-1, as indicated in Figure 1 and Figure 2. (**A**) N-fold change in cumulative cytokine production (Day 3 to Day 12 post-infection) of explants infected with HIV-1 BaL (squares, n = 5) and HIV-1 LAI (dots, n = 7) compared to donor-matched untreated explants (VEH). Lines indicate mean values. Asterisks denote a statistically significant difference with VEH (one sample *t* test,* *p* < 0.05, ** *p* < 0.01, *** *p* < 0.001). (**B**) N-fold change in cumulative cytokine production (Day 3 to Day 12 post-infection) of explants infected with HIV-1 BaL (squares, n = 1–2) and HIV-1 LAI (dots, n = 4) compared to donor-matched untreated explants (VEH). Lines indicate mean values. Asterisks denote a statistically significant difference with VEH (one sample *t* test,* *p* < 0.05, ** *p* < 0.01, *** *p* < 0.001). (**C**) N-fold change in cumulative cytokine production of explants treated with the indicated MPA concentration prior to infection (pre-infection, n = 12) or before and after infection (pre&after, n = 5–6) compared to donor-matched VEH. Lines indicate mean values. Asterisks denote a statistically significant difference between each treatment group and the respective RU-486-treated control (two-way ANOVA with Sidak’s multiple comparisons test, * *p* < 0.05, ** *p* < 0.01, *** *p* < 0.001).

**Figure 4 viruses-13-02303-f004:**
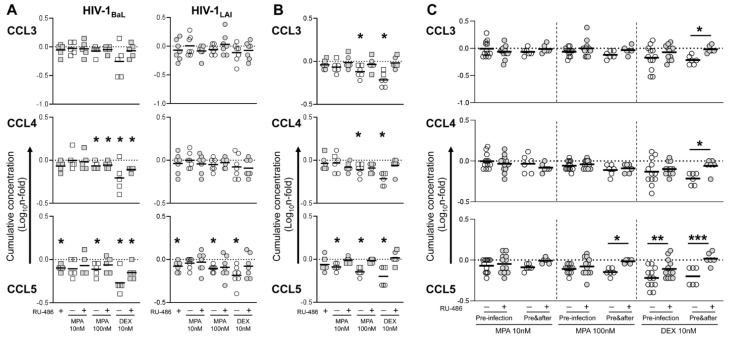
Cytokine production in HIV-infected tonsillar tissue explants treated with steroid hormones. Cytokine levels were measured in culture supernatant of tonsillar tissue explants treated with medroxyprogesterone acetate (MPA) and dexamethasone (DEX) at the indicated concentrations, in the presence or the absence of mifepristone (RU-486) at 1 μM, prior to infection (pre-infection, (**A**)), or before and after (pre&after (**B**)) infection with HIV-1, as indicated in Figure 1 and Figure 2. (**A**) N-fold change in cumulative cytokine production (Day 3 to Day 12 post-infection) of explants infected with HIV-1 BaL (squares, n = 5) and HIV-1 LAI (dots, n = 7) compared to donor-matched untreated explants (VEH). Lines indicate mean values. Asterisks denote a statistically significant difference with VEH (one sample *t* test,* *p* < 0.05, ** *p* < 0.01, *** *p* < 0.001). (**B**) N-fold change in cumulative cytokine production (Day 3 to Day 12 post-infection) of explants infected with HIV-1 BaL (squares, n = 1–2) and HIV-1 LAI (dots, n = 4) compared to donor-matched untreated explants (VEH). Lines indicate mean values. Asterisks denote a statistically significant difference with VEH (one sample *t* test,* *p* < 0.05, ** *p* < 0.01, *** *p* < 0.001). (**C**) N-fold change in cumulative cytokine production of explants treated with the indicated MPA concentration prior to infection (pre-infection, n = 12) or before and after infection (pre&after, n = 5–6) compared to donor-matched VEH. Lines indicate mean values. Asterisks denote a statistically significant difference between each treatment group and the respective RU-486-treated control (two-way ANOVA with Sidak’s multiple comparisons test, * *p* < 0.05, ** *p* < 0.01, *** *p* < 0.001).

## Data Availability

The findings of this study are available within this paper and its Appendix A.

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
