# Peer review of "The Progestin Medroxyprogesterone Acetate Affects HIV-1 Production in Human Lymphoid Tissue Explants in a Dose-Dependent and Glucocorticoid-like Fashion"

_viruses, 2021, doi:10.3390/v13112303_

Round 1
Reviewer 1 Report
The manuscript entitled "The progestin medroxyprogesterone acetate affects HIV-1 production in human lymphoid tissue explants in a dose-dependent and glucocorticoid-like fashion” by Vanpouille and co-authors focuses on the analysis of the effect of injectable progesterone-like contraceptive DMPA on HIV-1 infection and replication in lymphoid tissues. Impact of this compound on HIV-1 has earlier been studied mainly in the female genital mucosa and blood and has not been well documented in lymphoid tissue. Since lymphoid tissue is a major reservoir of HIV infection and both the progesterone (PR) and glucocorticoid receptors (GR) are present there, it is an important to understand how progestins such as MPA affect susceptibility of lymphoid cells to HIV-1 and how viral infection and replication can be regulated in these cells via synthetic agonist of these receptors, that initiates general anti-inflammatory effect. The authors investigated immunomodulatory effect of MPA using the explants of human tonsils, infected with HIV-1 ex vivo, and tested the viral p24Gag protein and major cytokines earlier shown responsible to HIV infection, in culture supernatant. The tissue specimens were collected from male and female children under 10 years of age, infected with HIV-1, and treated with MPA, dexamethasone (DEX) or progesterone (P4) with or without progesterone receptor antagonist RU-486 before or after the infection. The authors showed that MPA treatment, identically to DEX, reduced replication of both R5 and X4 HIV-1 strains (BaL and LAI respectively), as well as production of the pro-inflammatory cytokines IL-1β, IL-17A and CCL5, however significantly elevated CCL20 release, whereas P4 had no effect on HIV-1 replication. Overall, the authors demonstrated that increased MPA concentrations, comparable with those detected in the blood, affect the immune response in lymphoid tissue in a glucocorticoid-like manner and locally impact the HIV-1 replication. This result makes the manuscript relevant and important to understand a pharmacological effect of this widely used progesterone-like contraceptive on HIV infection dynamics.
Overall, the manuscript is of interest, well-written and logical, and is supposed to be important for an audience of virologists, immunologists and clinicians. The manuscript structure is clear and solid. In general, the study is technically correct, methodically rigorous, providing clear and precise results. Without diminishing the undoubted scientific value of this study, below I provide a few suggestions that might improve some points of the manuscript.
- In Introduction, the authors mentioned that the effect of the progesterone-like contraceptive DMPA on HIV-1 susceptibility and infection has earlier been studied in vivo. These studies demonstrated that “…high doses of P4 or MPA affect both the physical and immune defenses of the female genital mucosa (FGM) promoting the establishment of a productive infection with STIs, including HIV, simian immunodeficiency virus, and herpes simplex viruses”. In present work the authors showed decreased count of viral p24 protein in the media from HIV-1 infected tonsil explants treated with MPA and DEX. The issue is whether the experimental system used in this study is a representative model. The authors modelled the highest plasma concentrations of MPA in culture media. However, it is unclear whether the presenting of glucocorticoid receptor (GR) and especially progesterone receptor (PR) in children’s tonsils is the same as in tissues of adults, who are getting DMPA contraceptive on a regular base, and whether there are differences in this value between the male and female children. This analysis is not present in the manuscript, however the differences in PR expression/presentation may be responsible for the age- and sex-related discrepancies in the effect of MPA on HIV-1 infection. Despite GR is highly ubiquitous and presents in all tissues and organs, the potential age-related and sex-related differences should be discussed. While MPA affects cellular response mostly through the GR, an extra panel in Fig. 1 (or supplemental figure) showing MPA impact on HIV-1 separately in the tonsil explants from the male and female children could be added.
- The authors used two laboratory strains of HIV-1, BaL and LAI. Although the effects of MPA, P4, and DEX on their replication are described in the Abstract and then analyzed in the Results and Discussion sections, it is unclear why these two isolates were used. The authors did not mention that these HIV-1 strains have different cellular tropisms, which is probably the reason why they were selected for the study. This information should be added to the Abstract and subsection 3.2 of Results.
- Unlike other tested cytokines, CCL20 was shown upregulated in response to MPA and DEX. This is interesting result, especially in the context that DR agonists downregulate inflammatory genes and suppress inflammatory response. CCL20 is known as inflammation-related chemokine recruiting T cells and dendritic cells. Its elevated expression after DEX and MPA treatment might be implicated in the negative effect on the HIV-1 replication. This could be additionally discussed in the manuscript.
- Abstract, lines 20-21: “…the injectable contraceptive depot medroxyprogesterone acetate (DMPA) and HIV-1 susceptibility has been addressed…” Then only the acronym MPA is used. This should be clarified.
Author Response
We thank the reviewer for the positive feedback and valid remarks.
Point 1. In the revised manuscript and following the reviewer’s suggestion, we compared the effect of MPA, DEX and P4 on HIV-1 replication in females versus males. We carried out such comparison for the results from experimental infection setup 1 (pre-infection, which is shown in supplementary figure 2. As mentioned on line 212, we did not observe any difference in cumulative p24gag production in culture supernatant of explants treated with either MPA, DEX or P4 between female and male tissue donor.
As suggested by the reviewer, we also commented on the expression of PR and GR in lymphoid tissue and the potential relationship between their expression, sex and age. As reported on line 409, we cannot exclude that PR may be expressed by some cell types within tonsils, and/or vary under specific circumstances that are yet fully understood. While it is possible that some unknown factors specifically present in DMPA users in vivo may alter PR and GR functions in tonsils, thus limiting the relevance of our model, we demonstrated that tonsils from prepuberal children offer a robust experimental platform, and reproducible results were generated at two different study sites, although with different experimental design and infection setup. Therefore, we believe that our system could be further manipulated under controlled experimental conditions to make it more similar to the in vivo situation of DMPA users, without the interference of endogenous factors, such as sex hormones, which account for interindividual variability among women of reproductive age and would require extensive tissue donor characterization. On line 466, we also discuss the intrinsic limitations of our system owing to its narrow experimental time window that does not make it suitable to mirror the long-term effects of DMPA and those associated with a prolonged use.
Point 2. We highlighted this information in the abstract, on line 24, and explained the rationale of using HIV-1 variants with different cell tropism in the result section on line 190 of the revised manuscript. We also compared the effect of compound treatment on HIV-1 production between BaL- and LAI-infected explants as reported in supplementary figure 1 and in the discussion section on line 212. In the original version of the manuscript we commented on the effect of HIV-1 variant cell tropism on baseline cytokine production, now line 269.
Point 3. We discussed the implications of elevated CCL20 levels in the context of HIV-1 pathogenesis in the revised manuscript on line 432. Following the reviewer’s suggestion, we now add this point in the discussion and acknowledge the dual protective and detrimental function of this mediator of inflammation.
Point 4. In the revised manuscript we introduced the acronym MPA to indicate medroxyprogesterone acetate both in the abstract and introduction, on line 23 and 61 respectively.
Reviewer 2 Report
This study aims to address the effect of MPA on HIV-1 infection in lymphoid tissue using human tonsillar tissue explants. It is important to study the effects of MPA on HIV-1 infection in lymphoid tissue, as it is a major site of HIV-1 replication. However, several significant weaknesses need to be addressed to improve this manuscript.
- It is worrisome to draw a central conclusion that MPA reduces HIV replication by relying on a single assay throughout this study, especially for the conclusion contradicting many published results. The conclusion needs to be verified through different angles of HIV replication using several assays.
- The manuscript limits to the level of observing phenomenon without further elucidating the potential mechanism behind the phenomenon, which significantly reduces its impact in the field.
- The effect of MPA on the expression of the mentioned mediators of inflammation has been mostly published in the literature.
Author Response
We thank the reviewer his/her valuable comments.
Point 1. We do not see a contradiction between our study and others using different experimental models, i.e. PBMCs and cervical explants, as our work is the first to address the effect of MPA treatment on human lymphoid tissue in the context of ex vivo-HIV-1 infection. We speculated on the reason underlying the discrepancy between these reports in the original version of the manuscript, now lines 387-404. In the same paragraph, we also mentioned that a local reduction in HIV-1 replication associated with MPA treatment could be observed early upon challenge in a humanized mouse model of vaginal transmission, in spite of the fact that MPA-treated mice remained more susceptible to HIV, thus suggesting that the nature of MPA effect may depend on the time of analysis. This limitation of our model was acknowledged in the original version of the manuscript, now line 466.
Although we cannot exclude that sex hormone treatment could differently affect the release of HIV-1 p24gag and virus-associated RNA in culture supernatant, a strong positive correlation between p24gag concentration and RNA copy number in supernatant of ex vivo infected human tonsillar tissue explants was validated for both CCR5 and CXCR4 variants by the Margolis group, see reference 18 of the revised manuscript. Therefore we believe that the measurement of p24gag by itself offers a reliable tool to estimate novel virus production in our system.
Point 2. We partly agree with the reviewer. Our experiments were primarily designed to study the effect of MPA on HIV-1 replication in lymphoid tissue, which has not been previously reported. However, the comparison with multiple compounds with different affinity for PR and GR suggested that MPA elicits its immune-regulatory in a glucocorticoid-like fashion, thus providing a mechanistic insight. Future experiments will aim to validate the immunological correlates of MPA treatment in other tissues with potentially different PR expression patterns, such as the female genital mucosa, and identify the underlying cellular and molecular mechanisms, as discussed in the original version of the manuscript, now line 469.
Point 3. We respectfully disagree with the reviewer. The reports on the levels of most cytokines in samples from DMPA users and/or in vitro stimulated cells are inconsistent and vary with the time of sampling, donor characteristics and sample nature, as reviewed in references 8, 9 and 19 of the revised manuscript. As mentioned under point 2, this is the first study conducted in lymphoid tissue and, to the best of our knowledge, little to no data exist on the effect of MPA on either the systemic or local expression of CCL20 and IL-22 at this or other body sites, as discussed in the original version of the manuscript, now lines 430-460.
Reviewer 3 Report
In this work, Vanpouille and colleagues describe the effect of a commonly used injectable contraceptive, medroxyprogesterone acetate (MPA), on HIV-1 replication. The investigators use human tonsillar lymphoid tissue explants infected with two different HIV-1 strains, BaL and LAI, and measure HIV-1 replication and expression of various cytokines in cell culture supernatants using bead-based immunoassays. Treatments with dexamethasone, progesterone and RU-486, glucocorticoid receptor and progesterone receptor blocker, were used as controls. Results presented show a dosage dependent decrease in HIV-1 replication in MPA treated explants. Additionally, increasing the time of MPA treatment resulted in proportional decrease in virus replication. Dexamethasone treatment resulted in reduction of proinflammatory cytokines IL-1β and IL-17A, IL22, CCL4 and CCL5. Shorter MPA treatment resulted in a decrease of CCL5, and longer treatment affected expression of IL-1β, IL-17A, IL-22, CCL3, CCL4 and CCL5.
The topic is relevant considering the health impact and contradictory results found in previous studies and other model systems. The manuscript is well written, and experiments well controlled, but overall, in its current form, the study is rather preliminary and descriptive, and the scope of the study is rather narrow.
- The reported fold change in HIV-1 replication (0.7-fold decrease over vehicle treated control) is considered statistically significant, but not very convincing.
- Only bead-based immunoassay and multiplex cytokine immunoassays are used to determine virus titers and excreted cytokines, respectively. Other methods should be used to verify results.
- Sample size is rather small considering that other studies have been done to address the topic.
- Although using prepubescent tonsillar tissue overcomes the possible contribution of endogenous sex hormones, the relevance is somewhat unclear. In addition, in some of the samples the age of the donor was not known. A side-by-side experimental set up would help to address this issue.
Author Response
We thank the reviewer for the overall positive feedback and we acknowledge the preliminary nature of our study.
Point 1. Our experimental system was designed to achieve a productive infection and evaluate how compound treatment modulate virus production, in the absence of cell stimulation and/or selected growth factors. We described how explants were infected with different inocula to achieve comparable levels of virus replication between BaL and LAI on line 126. This does not necessarily reflect the natural route of infection of lymphoid tissue as it occurs in vivo. Nevertheless, explants remain a relevant platform to study the pathogenesis of HIV-1 and its modulation by endogenous and/or exogenous factors. Our results demonstrated a clear dose-dependent effect of MPA on HIV-1 as well as cytokine production, and reproducible across two different experimental infection settings (pre- vs pre&after infection) and study sites (NIH and Karolinska).
Point 2. We thank the reviewer and we will consider her/his suggestion to include additional analysis in our future studies. As mentioned in the response to reviewer 2 point 1, a strong positive correlation between p24gag concentration and RNA copy number in supernatant of ex vivo infected human tonsillar tissue explants has been previosuly reported for both CCR5 and CXCR4 variants by the Margolis group, see reference 18 of the revised manuscript. Therefore we believe that the measurement of p24gag by itself offers a reliable tool to estimate novel virus production in our system. On cytokine measurements, the bead-based multiplex immunoassay used in this work was validated and the analysis carried out in Dr. Margolis’ lab, which is part of the Microbicide Quality Assurance Program (MQAP) (see Fichorova R.N., et al. Biological and Technical Variables Affecting Immunoassay Recovery of Cytokines from Human Serum and Simulated Vaginal Fluid: A Multicenter Study. Analytical Chemistry 2008 80 (12), 4741-4751 DOI: 10.1021/ac702628q). We deem protein more informative than gene expression analysis in our study. The inclusion of additional analysis by in situ immunostaining and/or flow cytometry would allow the quantification and identification of cellular sources of selected cytokines that is of course of interest but beyond the scope of this work.
Point 3. Our study employs human tonsillar tissue as a model of HIV-1 pathogenesis in lymphoid tissue. All experiments were devised using donor-matched explants to account for inter-donor variability and multiple treatment groups within individual experiments. Although the sample size can be increased, we think that it was sufficient considering result reproducibility between experimental settings (pre- vs pre&after infection) and between the two study sites (NIH and Karolinska).
Point 4. Following the reviewer’s suggestion, in line with the request of reviewer 1 (see point 1), we compared the effect of MPA, DEX and P4 on HIV-1 replication in females versus males. We carried out such comparison for the results from experimental infection setup 1 (pre-infection), which is shown in supplementary figure 2. There was no difference in cumulative p24gag production in culture supernatant of explants treated with either MPA, DEX or P4 between female and male tissue donor. In the revised version of the manuscript we acknowledged the limitations of our experimental design, and discussed the potential effect of sex and age on GR and PR functions, see line 409.
Reviewer 4 Report
Introduction is missing some focus relative to genital/mucosal immunology and the impact of the studied compounds.....
Materials and Methods:
-Tonsils section needs details that could be taken from section 3.1...tonsillectomy from healthy individuals or?....how do you check for the donor's tonsil inflammatory status etc?
-elaborate on tonsil explant system.....assessment of cell death in the system?
-compounds....bind to which cell populations in explant?....perhaps reiterate the effects of compounds used in the study....toxicity?.....
Results
-3.2
Figure 1 is very heavy ....the protocol should be shown as "A " (idem for Figure 2)....
-3.3
-cytokine expression levels prior to HIV infection?.....
-day 3 to day 12....why not day 6 to day 12 as for HIV assessment....this should be clarified
-how do you check for selected depletion or expansion of a cell population in your system.....this should be clarified
-production in supernatant.....verify on tissue by IHC, ISH?
-direct impact of compounds on isolated cell populations
Figure 3 are Figure 4.....cytokines and chemokines should be grouped, and the title of these Figures should be accordingly......with appropriate legends
Discussion
limitations of the system should be discussed
conclusions are overreached.....some statements should be nuanced
Author Response
We thank the reviewer for her/his valuable comments. However, some remarks are incomplete. We grouped comments under 4 main points and responded accordingly.
Please see the attachment.

Round 2
Reviewer 2 Report
Thanks for the authors’ effort to respond to the reviewer’s comments. However, the authors did not provide essential experimental evidence to prove the influences of MPA on HIV release and connect the dots between MPA-induced immune-regulatory and HIV production. Therefore, the reviewer’s major concerns have been addressed. The reviewer thinks the manuscript in the current version is not ready for the journal of Viruses.
Reviewer 3 Report
In this revised manuscript, Vanpouille et al respond to all of the three reviewer’s comments and address most of the concerns raised. Experiments contain appropriate controls, and although the scope of the study is still rather narrow and descriptive, it still merits publication due to the conflicting past results and high impact on health.